# Identification of the suprachiasmatic nucleus venous portal system in the mammalian brain

Yifan Yao [1], Alana B'nai Taub [1], Joseph LeSauter[2] & Rae Silver [1,2,3✉]

There is only one known portal system in the mammalian brain - that of the pituitary gland, first identified in 1933 by Popa and Fielding. Here we describe a second portal pathway in the mouse linking the capillary vessels of the brain's clock suprachiasmatic nucleus (SCN) to those of the organum vasculosum of the lamina terminalis (OVLT), a circumventricular organ. The localized blood vessels of portal pathways enable small amounts of important secretions to reach their specialized targets in high concentrations without dilution in the general circulatory system. These brain clock portal vessels point to an entirely new route and targets for secreted SCN signals, and potentially restructures our understanding of brain communication pathways.

[1] Columbia University Department of Psychology, 1190 Amsterdam Avenue, New York City, NY 10027, USA. [2] Department of Neuroscience, Barnard College, 3009 Broadway, New York City, NY 10027, USA. [3] Department of Pathology and Cell Biology, Graduate School, Columbia University Medical School, New York City, NY 10032, USA. ✉email: Rae.Silver@columbia.edu

The great majority of capillary beds drain into the veins of the general circulatory system, which then drain into the heart, rather than into another capillary bed. The only established exception to this pattern in mammalian brain is the hypothalamic-pituitary portal system, first demonstrated over 90 years ago[1]. Brain portal pathways have been very difficult to study. While it is known today that the pituitary portal pathway is essential for survival and reproduction, it took about two decades to establish the direction of blood flow[2]. About three decades thereafter, the identification of neurohormones that travel in this portal system led to Guillemin and Schally's 1977 Nobel prize in physiology and medicine. Today, the concept of an "endocrine hypothalamus" is well established, but how these neurons communicate their secretions is not fully known. The discovery of additional portal systems in the brain would launch a bonanza of new directions into research on the neurovasculature, a requirement for understanding connectomes in the brain[3].

Like other hypothalamic neurons, those of the suprachiasmatic nucleus (SCN) produce neurosecretions. The SCN is the locus of the brain's circadian clock[4,5]. Several lines of evidence indicate that the SCN produces functional diffusible output signals. Transplants of SCN tissue rescue circadian rhythms of locomotor activity in arrhythmic SCN-lesioned host animals[6] with the period of the donor animal[7], irrespective of the attachment site within the 3rd ventricle (3V)[8]. SCN transplants are effective even when the grafted tissue is encapsulated in a polymer plastic that blocks fiber outgrowth[9]. Signals that diffuse from the SCN include paracrine outputs such as transforming growth factor alpha[10], prokineticin2[11,12] and cardiotrophin-like cytokine[13] and the peptides vasoactive intestinal polypeptide, arginine vasopressin (AVP), and gastrin releasing peptide[14,15].

Despite substantial evidence of effective humoral signals of SCN origin, a challenge regarding their biological significance is how this very small nucleus of about 20,000 neurons could possibly produce sufficient product to orchestrate rhythms throughout the body, unless the target of those secretions lies very nearby[16]. Here we identify a vascular pathway in mouse for communication of diffusible signals in a hypothalamic portal system connecting the SCN and a nearby circumventricular organ (CVO), namely the organum vasculosum of the lamina terminalis (OVLT), using tissue clearing methodology combined with multilabel immunostaining and light-sheet microscopy. The findings pave the way for using currently available tools to explore additional portal systems between other CVOs and nearby brain regions.

## Results

**Identification and visualization of the SCN, OVLT, and capillaries.** The vasculature of CVOs in vertebrates, including humans have long been of interest and have been investigated in substantial detail[17–20], while that of the SCN is less well established[17,21]. Our protocol enables very detailed visualization of the SCN-OVLT vasculature. To provide the most complete dataset and to enable mapping of the present results to publications using previously available techniques, the results are presented in three brain orientations (horizontal, sagittal, and coronal). Additionally, because blood vessels tend to course in tortuous planes, we provide three dimensional (3D) views and maximal intensity projection (MIP) images. The MIPs project the voxel with the highest attenuation value in every view throughout the 3D volume onto a 2D image, and thus provide sharper views of the complex course of portal capillaries of interest at optimal planes and depths. To augment the impression of depth, we also present a 3D animation in a movie (Supplementary Movie 1). Finally, schematics are provided to interpret the micrographs and

to indicate the relation of the pituitary portal system to the SCN-OVLT portal system.

Because the blood vessels that course between the SCN and OVLT lie along the midline on the very floor of the 3V, they are readily destroyed during physical sectioning and tissue processing for anatomical analyses. To retain the integrity of these blood vessels, we prepared whole mounts of the mouse brain, from the olfactory tubercle to the medulla using an iDISCO protocol. To identify the SCN, OVLT, and the capillaries connecting these structures, we performed triple label immunochemistry using AVP to delineate the SCN[22], collagen as a universal blood vessel marker[23], and smooth muscle actin (SMA) as an arterial marker[24], and performed high power scanning using light-sheet microscopy. A schematic showing the methods pipeline is given in Supplementary Fig. 1.

A ventral view of the region from the olfactory tubercle to medulla captures the OVLT and another CVO, the median eminence (ME), the AVP-stained SCN, and the midline blood vessels lying between these CVOs (Fig. 1). Because the SCN and OVLT were identified with specific markers and included in a single microscopic scan, it was not necessary to register these brain regions of interest against a standard atlas and the images didn't require stitching. Furthermore, the iDISCO method used produced relatively little shrinkage of about 11%[25], allowing estimation of the distance between the SCN and OVLT.

The sagittal and horizontal scans of the SCN, OVLT, and the capillaries lying between these nuclei are shown in Fig. 2 and Supplementary Fig. 3 respectively. Both orientations provide clear evidence of portal vessels connecting these structures. Each SCN has a pear-shaped main body and directly from the medial-most aspect, two prongs protrude rostrally (termed SCN rostrum, SCNr; also seen in Supplementary Fig. 2, AVP panel). The traces highlight several vessels emerging from the SCNr to form a portal system running close to the midline and reaching the ventral OVLT. In the 3D views of Fig. 2a, a depth of ~730 μm is shown. The attachment point of the portal system to the SCN lies at the rostral-most tip of the nucleus, where the left and right prongs of the SCN connect (Supplementary Fig. 3e). The same scans are also shown in maximum intensity projections (MIPs), at a shallower depth (620 μm near SCN, 350 μm between the SCN and OVLT, 410 μm near OVLT) to enhance the visualization of the portal capillaries (Fig. 2c). The results clearly show the two distinct compartments of the OVLT, namely the deep and superficial plexus. Interestingly, the portal vessels connect to the ventral most aspect of the superficial plexus of the OVLT as seen in a high magnification view (Fig. 2e).

We next sought to visualize the portal vessels and how they coursed in the 3V floor (3VF) in greater detail. To this end, we made 3D and MIP scans in higher power views of horizontal and coronal orientations in material triple labelled for collagen, SMA and AVP, and in traced capillaries. The results unequivocally show the continuity of the capillary portal vessels coursing between the SCN and the OVLT. Details of the horizontal and coronal scans of the SCN, OVLT, and the capillaries lying between these nuclei are shown in Fig. 3 and Supplementary Fig. 4, respectively. For each figure, a triple labelled image of the SCN-OVLT region is shown followed by a merged image of traced capillaries with immunostaining, and finally an explanatory schematic. For a very fine-grained analysis of the midline blood vessels, we made serial optical slices in horizontal (Fig. 3c, in 50 μm steps) and coronal (Supplementary Fig. 4b, in 100 μm steps) orientations, from the SCN to the OVLT to show the continuity and route taken by the portal capillaries. The main body of the two SCN nuclei are connected by capillaries that lie in the 3VF above the optic chiasm (OC) thus sharing a common blood supply (Horizontal view, Fig. 3ci–v; Coronal view,

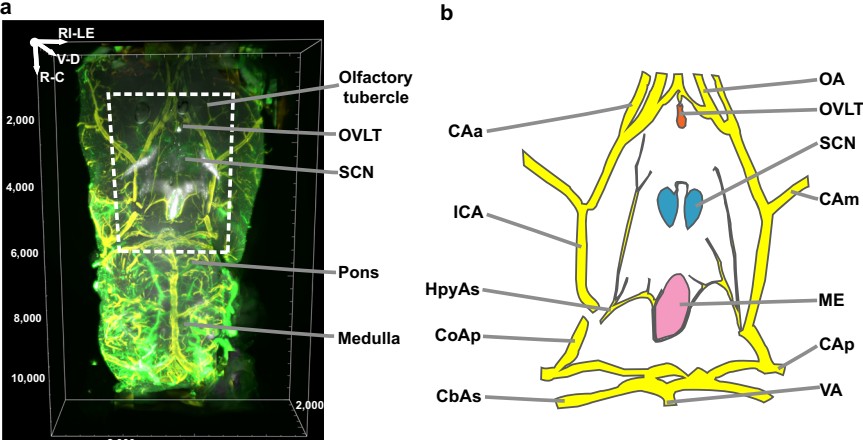

**Fig. 1 Ventral aspect of mouse brain. a** Ventral view of the area from the olfactory tubercle to medulla in iDISCO cleared tissue. The scanned volume is from 2.10 mm (Bregma) to −7.9 mm (rostrocaudally) and ~3 mm from the midline on each side. Arginine vasopressin (AVP) = white, collagen = green, and smooth muscle actin (SMA) = yellow. **b** Line drawing shows structures within the dashed frame shown in Fig. 1a. Circle of Willis = yellow, OVLT = orange, ME = pink and SCN = blue. CAa anterior cerebral artery, CAm medial cerebral artery, CAp posterior cerebral artery, CbAs superior cerebellar artery, CoAp posterior communicating artery, HpyAs hypophyseal artery superior branch, ICA internal carotid artery, ME median eminence, OA olfactory artery, OVLT organum vasculosum of the lamina terminalis, SCN suprachiasmatic nucleus, VA vertebral artery. Reference axis denotes the orientation of the tissue; R rostral, C caudal, V ventral, D dorsal, RI right, LE left. Scale unit = μm unless otherwise specified.

Supplementary Fig. 4bi–iv). Additionally, capillaries form anastomoses along the SCNr (Horizontal, Fig. 3bi–iii, 3cvi–viii; Coronal, Supplementary Fig. 4bv). Note that the portal vessels lie right near the midline, along the 3VF above the optic chiasm (Horizontal, Fig. 3cix–xi; Coronal, Supplementary Fig. 4bvi–vii). More laterally lying vessels do not contribute to this portal system.

Overall, the integrity of the tissue is confirmed by intact blood vessels and excellent correspondence with prior studies on the orientation and relative size of the hypothalamic region and ME, OVLT, and SCN: rostrocaudally ~560 μm for SCN main body and ~160 μm for SCNr; dorsoventrally ~350 μm; mediolaterally ~300 μm; distance of ~385 μm between OVLT and SCN (Fig. 3a). The OVLT lies rostral to the ME and pituitary gland at a distance of ~550μm[22,26–29].

**Comparison of SCN shell and core vasculature.** The core and shell of the SCN have different functions in orchestrating rhythmicity. The core receives retinal input from the retinohypothalamic tract, and is important in synchronizing the clock to the local environment. Core neurons tend to project to the shell, the locus of AVP-expressing cells. AVP is a major output and synchronization signal of the SCN[30,31], though there is abundant evidence of many other potential output signals[10–13]. Images of optical slices of cleared tissue (Fig. 4) and confocal microscopic images of immunochemically stained material in three orientations through the mid-SCN suggest a denser and more complex vascular network in the SCN shell (SCNs) than in the SCN core (SCNc) (Supplementary Fig. 5). This is confirmed quantitatively in tracings of collagen labeled vasculature (Fig. 4bi, ii; Supplementary Fig. 5c) and by the density of branch nodes in the SCNc and SCNs (Fig. 4biii). The greater complexity of the capillary network in the SCNs is consistent with its role in producing output signals.

Our calculations of branch point density are consistent with the study of whole-brain vasculature by Kirst et al.[24]. The collagen marker we used is a component of all blood vessels[23], while Kirst et. al used a cocktail of three antibodies[24]. To compare our results to theirs we calculated the branch point density of the whole SCN, and found that our results (0.132 ± 0.008 per unit

volume) were very similar to theirs (0.138 ± 0.029 per unit volume).

**Relation of SCN-OVLT to Hypophysial portal system.** The major features of the SCN-OVLT and the hypophyseal-pituitary portal systems are shown in Fig. 5. The sagittal view emphasizes the proximity of the SCN and OVLT, important for effective communication of diffusible signals produced by the very small SCN nucleus[16]. Capillaries branching extensively in the AVP-rich shell region emerge from the bridge of the SCNr, and traverse the 3VF to reach the very dense capillary plexus of the OVLT. Hypothalamic neurosecretory neurons travel to the ME, another CVO. The portal vessels lying along the pars tuberalis join the capillary bed of the ME to the anterior lobe of pituitary gland via the portal vessels.

**Discussion**

The present findings demonstrate a portal pathway connecting the capillary beds of the central brain clock in the SCN and the OVLT. The results suggest a solution to the enigma in neuroscience on how small populations of neurons in the brain can produce effective concentrations of secretions into the vascular system. In the specific case of the SCN brain clock, while the efficacy of diffusible signals is well established, it is unknown however how this small nucleus, comprised of ~20,000 small neurons (<10 μm) can generate a concentration of signaling proteins in the body fluids that is sufficient to activate their cognate receptors. Evidence from SCN transplant studies[32] and from calculations on the distance that diffusible SCN neurosecretions could travel[16], suggest that the humoral signals have to act on a nearby target. The OVLT, lying at a distance of 385 μm from the SCN meets the criterion.

**SCN-OVLT communication.** The OVLT has been implicated in numerous centrally regulated processes, including anticipatory activity, fever, sickness behavior, gonadal function, and systemic osmoregulation[33]. Each of these functions is under circadian control: anticipatory thirst[34]; ovulation[35]; osmoregulation[34,36,37]; fever and sickness behavior[38–40]. The OVLT is a sensory CVO that features a fenestrated vasculature and enriched receptors for

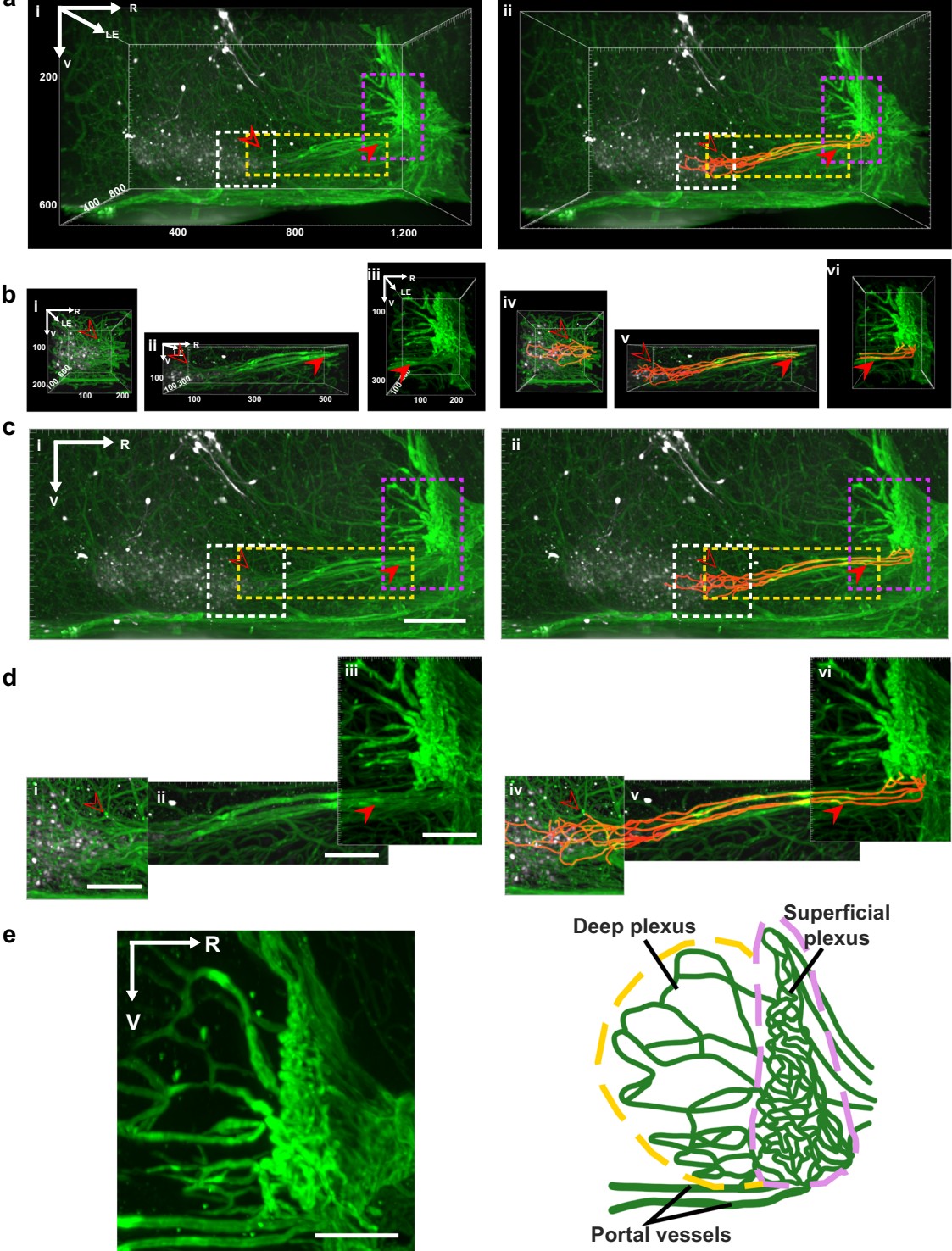

**Fig. 2 Sagittal view of portal vasculature between the SCN and OVLT in 3D and maximum intensity projection images. a–d** The left panels show capillaries coursing between the SCN and OVLT and the right panels show the same panels with traced vessels. **a, b** 3D view of vessels between SCN and OVLT at low (**a**) and higher (**b**) magnification. Areas outlined in the boxes in (**a**) are shown at higher magnification in (**b**) for the regions (left to right) near the SCN, between the SCN and OVLT, and near OVLT. **c, d** Maximum intensity projection of data shown in Fig. 2a, b respectively at low (**c**) and high magnification (**d**). This view reduces the depth of the scan and enhances visualization of some of the portal vessels coursing between the SCN and OVLT (N = 8 mice). **e** Maximum intensity projection of an optical slice (left, 100 μm) and accompanying schematic (right) highlight the entry points of portal vessels into the ventral superficial plexus of the OVLT. (N = 8 mice). Legend details: In panels **a–d**, the open and closed arrows point to a specific vessel lying at the rostral SCN or near the ventral OVLT respectively. Immunostaining labels as in Fig. 1a and orange = tracings of capillaries. For **a–b**, volume of images is shown in the axes. For **c–d**, maximum intensity projections depth: 620 μm (i, iv), 350 μm (ii, v), 410 μm (iii, vi). Reference axes: R rostral, LE left; V ventral. Scale bar = 100 μm.

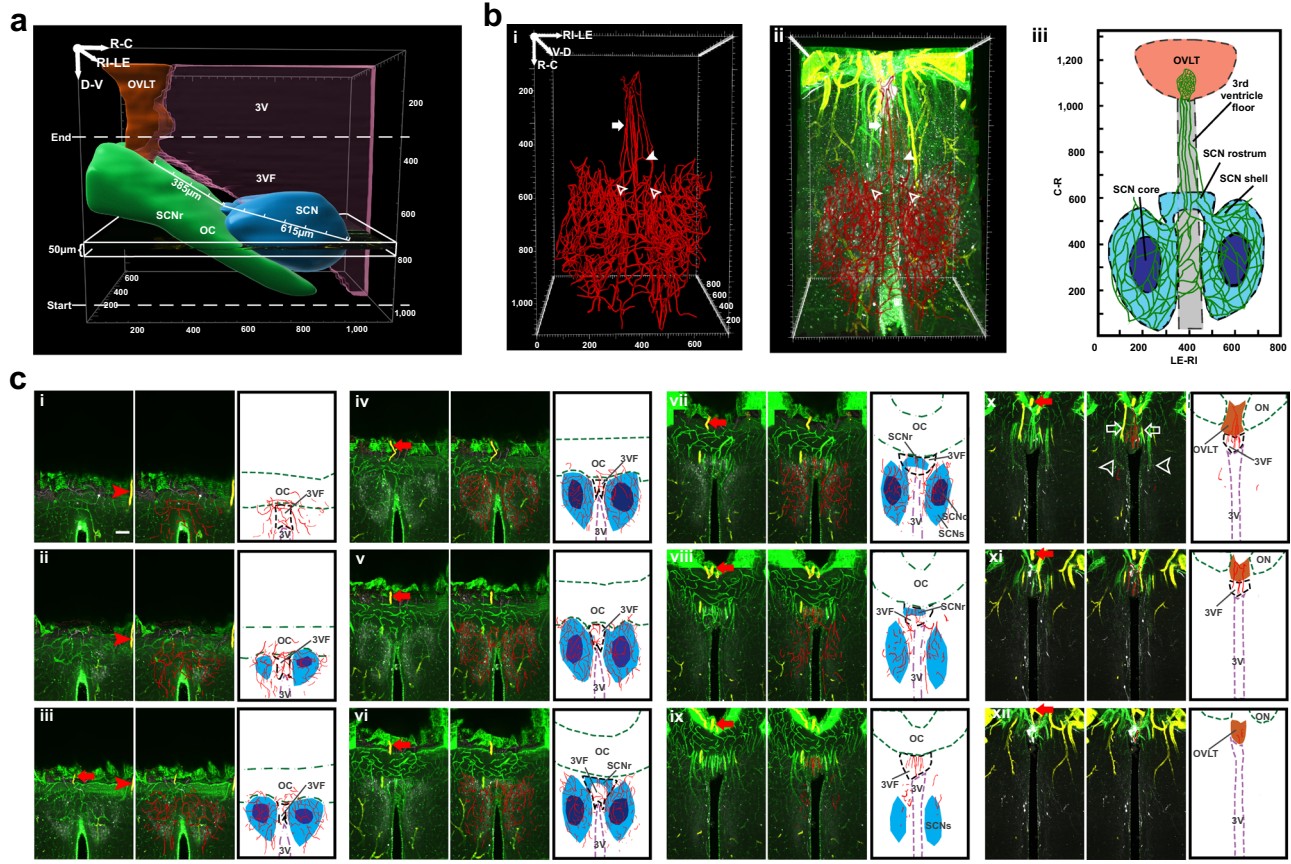

**Fig. 3 Horizontal view of capillaries connecting SCN and OVLT. a** The white rectangle shows the orientation of the horizontal serial scans, starting from the base of the brain. **b** Traced blood vessels connect the SCN and OVLT. (N = 6 mice). **bi** Vasculature reconstruction from traces of blood vessels connecting SCN and OVLT. **bii** Merged AVP, collagen, SMA, and traced blood vessels. Capillaries emerge independently from the SCNs left and right side (open arrowheads). These form anastomoses in the SCNr (solid arrowhead). Branches exiting from SCNr (solid arrow) course along the 3VF before joining the OVLT capillary bed. **biii** Explanatory drawing. **c** Serial optic slices (50 µm) of vasculature between the SCN and OVLT demonstrating continuity of portal capillaries (N = 6 mice). The plates show triplets of images as follows: Left panel = merged AVP (white), collagen (green) and SMA (yellow); red arrows and arrowheads are place markers indicating landmarks for orientation in adjacent slices. Middle panel = blood vessel traces are superimposed on immunochemical results of the left panel; Right panel = drawing identifying structures in the middle panel. Details of the serial plates are as follows: **ci–v** The left and right SCN capillary beds form anastomoses via capillaries traveling in the 3VF. **cvi–viii** The traced blood vessels in the SCNr form anastomoses. **cix–xi** The portal blood vessels (red traces, middle panel) lie at the midline. **cx** Veins that are not part of the portal system (middle column, white arrowheads) and arteries (white arrows) lie more laterally. **cxi–xii** The midline blood vessels enter OVLT from its ventral aspect. Abbreviations and colour codes: 3VF floor of the third ventricle, SCNc SCN core, SCNr SCN rostrum, SCNs SCN shell, remaining abbreviations as in Fig. 1. OVLT orange, SCNs and SCNr light blue, SCNc dark blue, 3V magenta, and OC green. Scale bar = 100 µm.

hormones and neuropeptides[41]. The weight of this evidence implies that the SCN imposes circadian changes on functions of the OVLT and suggests but does not prove that the direction of communication is from the brain clock to the OVLT. Also, it is established that the efferent projections from the SCN reach the OVLT indicating dual neural and haemal control. Presumably neural communications support fast acute signals while humoral signals are slower and support long-term regulatory functions. Multiple such temporal aspects controlling water balance and thirst have been reported (reviewed by Zimmerman[42]).

**Neurosecretions.** An important question is the identity of neurosecretions that travel in the SCN-OVLT portal pathway. The best-characterized output signal of the SCN is the peptide AVP[43–45]. AVP released from the SCN controls the daily hormonal rhythm in hypothalamic-pituitary-adrenal and in hypothalamic-pituitary-gonadal axes[45]. AVP is also reported to control rhythms in locomotor activity[46,47], and nesting behavior[48]. Importantly, peptides generally and AVP specifically,

can enter parenchymal capillaries, including those bearing tight junctions[49–51]. However, AVP is but one possibility, as the SCN bears large numbers of neuronal antigens[52–54].

**Question of additional brain portal systems.** In the decades since the identification of the pituitary portal system, the occurrence of portal systems between CVO's and parenchyma have previously been considered but never unequivocably demonstrated. Szabó[20] discussed a vascular connection among three CVOs namely the choroid plexus, subfornical organ, and the OVLT and wondered about "neuro-haemal connections." He suggested that the superficial network of the OVLT is interconnected with nearby hypothalamic regions (the preoptic region, subependymal preoptic recess, and the retrochiasmatic area) but did not characterize the vessels as capillaries.

Based on morphological data, Grafe and Weindl[55] argued against a portal circulation between OVLT and the preoptic area. Cammermeyer[56] examined the area postrema and noted a connection between the "wide" and "narrow" capillaries, but

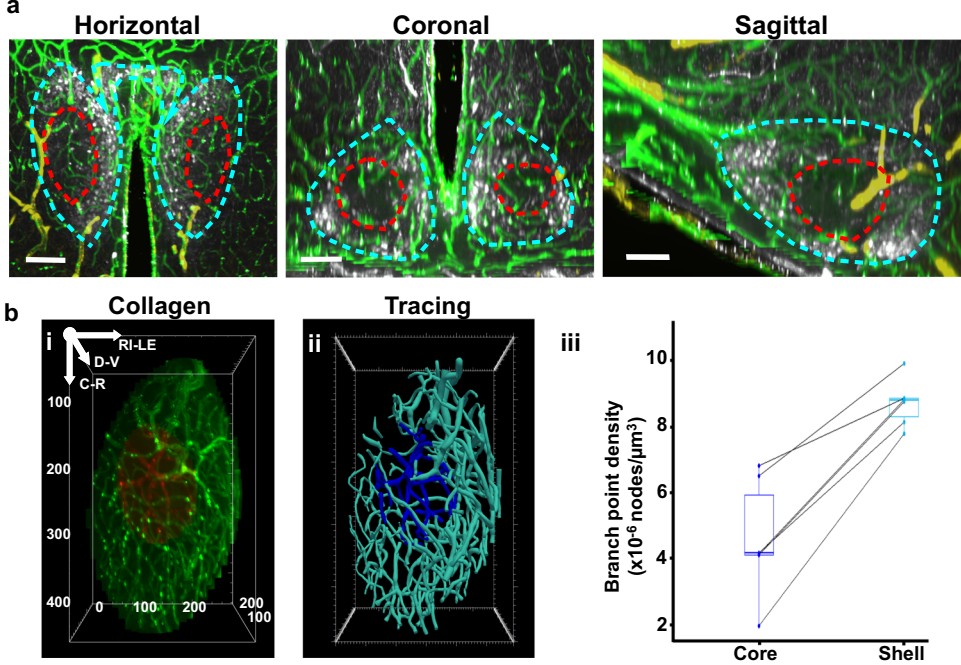

**Fig. 4 Capillary network of SCN core vs. shell. a** Optical slices (50 μm) of a representative SCN in horizontal, coronal, and sagittal orientation indicate a denser and more complex vasculature in the AVP-abundant shell (dashed blue line) vs. core (dashed red line) ($N = 6$ mice). Scale bar = 100 μm. **b** Core and shell SCN vasculature network. **bi** Collagen staining of SCN core (red) and shell (green) blood vessels. **bii** Tracings of SCN core (dark blue) and shell (light blue) blood vessels. **biii** Comparison of core and shell branch point density ($N = 6$ SCN). SCN core (dark blue box) vs shell (light blue box): $4.6 \pm 1.8 \times 10^{-6}$ vs. $8.7 \pm 0.7 \times 10^{-6}$ nodes/μm$^3$; two-tailed paired $t$ test, $t(5) = 7.81$, $p = 0.00055$. The center line of the boxplot refers to median; the upper box limit refers to 75th quantile and the lower limit refers to 25th quartile; whiskers represent 1.5x interquartile range; points beyond whiskers are outliers. A single line connects dots representing measurements of the core and shell branch points from the same SCN.

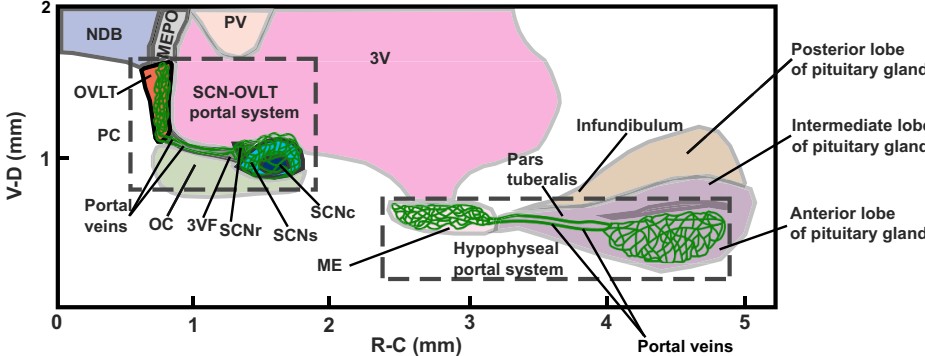

**Fig. 5 Relation of SCN-OVLT portal system to the hypophyseal portal system.** The schema shows a sagittal view of the SCN-OVLT and hypophyseal portal systems. MEPO median preoptic nucleus, NDB nucleus of diagonal band, PV periventricular hypothalamic nucleus, PC preoptic cistern; remaining abbreviations as in Figs. 1, 3. Scale unit = mm.

concluded that "no portal circulation was found." In contrast, Roth and Yamamoto[57] suggested that a portal system is present but cautioned that "The number and tortuosity of the capillary bed increased rostrally (Fig. 2 a–e), making it exceedingly difficult to trace any single vessel through several sections."

CVOs are highly conserved in from teleosts to mammals[58] and the results in mouse are likely to be in other mammals as well. That said, it remains to be determined if other CVOs bear portal systems that participate in local communications between the brain, blood, and cerebrospinal fluid. Late 19th century metaphors of the nervous system envisioned it as a syncitium on the one hand (Golgi) and as discrete neurons on the other. While Ramon y Cajal won that battle, a parallel metaphor suggested by the present results, however, push us to consider

that both the images are apt and that the vascular network of the brain is a key to understanding communication systems of the brain.

## Methods

**Animals**. Male C57BL/6NJ mice bought (Jackson Laboratory, Bar Harbor, ME) at age 8 weeks, were adapted to the lab for at least 2 weeks prior to the start of the study. Mice were provided with *ad libitum* access to food and water, and maintained in a 12:12 h light:dark (LD) cycle. The room was maintained at 21 ± 2 °C and 35–70% humidity. Five hours after lights on, animals were deeply anesthetized with ketamine (100 mg/kg) and xylazine (10 mg/kg) and perfused intracardially with 50 ml 0.9% saline followed by 100 ml 4% paraformaldehyde (PFA) in 0.1 M phosphate buffer (PB) at pH 7.3. After post-fixing in 4% PFA overnight, brains harvested for whole-brain iDISCO clearing ($N = 27$) were transferred to 0.1 M PB with 0.9% saline (PBS). Brains to be imaged with confocal microscopy ($N = 3$) were cryoprotected in 20% sucrose in PBS for at least 48 h. All experiments were

performed according to protocols approved by the Institutional Animal Care and Use Committee of the Columbia University in accordance with guidelines set by the NIH (Protocol AC-AABH1603).

**iDISCO protocol.** Brains were trimmed to include the region from the olfactory tubercle to the medulla in the rostro-caudal plane, from the paraventricular nucleus to the base of the brain with maxilla attached in the dorso-ventral plane and 3 mm from the midline on both sides in the medio-lateral plane. The iDISCO protocol used in this study (which produces tissue shrinkage of ~11%), is based on Renier et al.[25] The fixed tissues were washed in PBS for 30 min (3x). The tissues were then dehydrated with methanol at serially increasing concentrations for at least 1 h each (20, 40, 80, and 100%) and washed with fresh 100% methanol for another hour. The tissues were then transferred to 66% dichloromethane (DCM) /33% methanol with shaking overnight to remove lipids. The next day, tissues were washed twice in 100% methanol and then transferred to 5% hydrogen peroxide in methanol to be bleached overnight and on the following day, the tissues were rehydrated with methanol at serially decreasing concentrations (80, 60, 40, and 20%) and then washed twice in PBS with 0.2% TritonX-100 (PTx.2). Next, the tissues were incubated in permeabilization solution [2.3% (w/v) glycine and 20%(v/v) dimethyl sulfoxide (DMSO) in PTx.2] for 2 days at 37 °C, transferred to blocking solution (10% DMSO and 6% donkey serum in PTx.2) and incubated for 2 days. After blocking, the tissue was incubated in the primary antibodies for two weeks and then washed 4–5 times in PBS with 0.2% (v/v) Tween-20 and 0.01% heparin (w/v) (PTwH) until the next day and incubated in the secondary antibodies. After 1 week of incubation in the secondary antibodies, tissues were washed with PTwH for 1 day. Next the tissues were dehydrated with methanol series again and incubated in 66%DCM/33% methanol for 3 h. Finally, the tissues were washed with DCM twice and transferred to benzyl ether until completely cleared.

**Preparation of sections.** Brains were sectioned (50 μm) on a cryostat (Microm HM 500 M, Walldorf, Germany). Free-floating sections were washed in PBS with 0.1% Triton X-100 (0.1% PBST) and incubated in 1:100 normal donkey serum in 0.3% PBST for 1 hour before immunostaining. Slices incubated with primary antibodies were washed with 0.1% PBST and then incubated with secondary antibodies. Immunostained slices were washed with PB and mounted in PBS on subbed slides and coverslipped with Fluoromount Aqueous Mounting Medium (Sigma-Aldrich, F4680, St. Louis, MO) and cover glass No. 1 (Fisher Scientific, 12-544-18, Waltham, MA).

**Antibodies.** The following primary and secondary antibodies were used: anti-AVP (rabbit, ImmunoStar, 20069, Hudson, WI); antitype IV collagen (goat, SouthernBiotech, 1340-01, Birmingham, AL); anti-SMA (1:67, mouse, Dako, M0851, Santa Clara, CA); donkey antirabbit Cy2 (Jackson ImmunoResearch, 711-225-152, West Grove, PA); donkey antirabbit Cy3 (Jackson ImmunoResearch, 711-165-152); donkey antimouse Cy3 (Jackson ImmunoResearch, 715-165-151); donkey antigoat Cy5 (Jackson ImmunoResearch, 705-175-147); tomato lectin-fluorescein (Vector laboratories, FL-1171, Burlingame, CA).

**Immunostaining.** For the iDISCO protocol, optimization for double- and triple-labelling parameters was performed. For the double-labelling protocols ($N = 8$), the following dilutions were tested: anti-AVP at 1:10,000, 1:5,000, 1:2,500, and no-primary control with 1:200 donkey antirabbit Cy2; antitype IV collagen at 1:250, 1:125, 1:50, and no-primary control with 1:200 donkey antirabbit Cy5. The best signal-to-noise ratio was anti-AVP at 1:5,000 and antitype IV collagen was 1:125. For the triple-labelling procedure, a dilution series for SMA ($N = 16$) was determined using anti-AVP at 1:5,000 with donkey antirabbit Cy2 at 1:200; antitype IV collagen 1:125 with donkey antigoat Cy5 at 1:200, anti-SMA at 1:200, 1:100, 1:67 with donkey antimouse Cy3 at 1:200; no-primary controls. The optimal concentration for anti-SMA was 1:67. Additional animals ($N = 3$) were run at the optimal triple label primary concentrations (anti-AVP at 1:5000, antitype IV collagen at 1:125, anti-SMA at 1:67). There was no staining in the absence of primary in any condition. Brain slices to be imaged with confocal microscopy ($N = 3$, one for each in horizontal, coronal, and sagittal orientation) were incubated at room temperature (RT) on a shaker with anti-AVP at 1:5000 for 1 h and 36 h at 4 °C and tomato lectin-fluorescein for 2.5 h at RT and for 1h at 4 °C.

**Light-sheet microscopy.** Cleared, immunostained tissue was imaged using the Ultramicroscope II (LaVision BioTec, Bielefeld, Germany) equipped with a LaVision BioTec Laser Module and an Andor Neo sCMOS camera with a pixel size of 6.5 μm, and acquired with ImSpector (version 7.0.119, LaVision BioTec). The filter sets used for excitation were: for AVP-Cy2, 488 nm, for SMA-Cy3, 525/50 nm, and for Collagen-Cy5 639 nm diode laser. The filter sets used for detecting light emission were: 525/50 nm for AVP-Cy2; 605/52 nm for SMA-Cy3; 705/72 nm for Collagen-Cy5.

The brain was mounted for either sagittal or horizontal imaging. Low power images were taken with a 0.1 NA/9.0 mm WD LaVision LVMF-Fluor multi-immersion objective with a 5 μm step size and a field of view (FOV) of 10.8 mm × 12.8 mm. The scan covered brain regions from the olfactory tubercle to the medulla, ~3 mm from midline in both hemispheres, generating a volume of 6,400 μm × 2,500 μm x 9,700 μm (left-right × ventral-dorsal × rostral-caudal) scanned horizontally (Fig. 1).

Higher power images were taken with a 12 × 0.53 NA/9.0 mm WD LaVision PLAN xDISCO objective using a 2 μm step size and a FOV of 1.39 mm × 1.17 mm, and included the region from the OVLT to the retrochiasmatic area. This generated a volume of 960 μm × 640 μm × 1390 μm (left-right × ventro-dorsal × rostral-caudal) for the sagittal scan (Fig. 2; horizontal view of the same sample is shown in Supplementary Fig. 3) and 730 μm × 1000 μm × 1120 μm (left-right × ventro-dorsal × rostral-caudal) for the horizontal scan (Fig. 3; coronal optical sectioning of the same sample is in Supplementary Fig. 4). As the FOV of the camera is large enough to capture the dimensions of the area of interest, stitching was not required in the current study.

**Image processing.** The images of cleared tissue were imported into Imaris (version 9.5.1, Bitplane AG, Zurich, Switzerland). The masking tool in the "Surface" module was used to reveal the spatial relationships among SCN, OVLT, 3V, and OC (Fig. 3a and Supplementary Fig. 4a). The SCN was identified by immunostaining for AVP[22] and the OVLT and OC were delineated in the collagen-labelled material.

Because the region of interest (ROI) was identified by immunostaining in all experiments, registration of the brain to a standard was not required.

The "Surface" module was also used to simultaneously visualize regions bearing high and low expression levels of proteins (e.g. Supplementary Fig. 2), as follows: In the AVP channel, to better detect the SCN, the PVN was masked. In the collagen channel, the pial signal of the ventral OC was masked. A new channel with enhanced SCN signal was merged with the channel devoid of the OC signal. In the SMA channel, to better display the blood vessels in the maximum projection, the signal from the ventral optic chiasm was masked. The SCN shell and core were delineated by the presence or absence of AVP respectively (Supplementary Fig. 5b).

**Vascular tracing and optic slicing.** Capillary vessels coursing between SCN and OVLT were traced by Imaris (Fig. 2 and Supplementary Fig. 3) or Vesselucida 360 (version 2019.1.3, MicroBrightField, Williston, VT; Fig. 3, 4 and Supplementary Fig. 4). Masked SCN core and shell with the collagen-labelled vasculature (Fig. 4bi) was exported to Vesselucida 360 for tracing. The tracing with Imaris was done manually using the "Auto-depth" method. Tracing with Vesselucida 360 was done using the semi-manual user-guided tracing with a directional kernel method for optimal sensitivity and accuracy in detecting blood vessels. Serial sectioning generated from horizontally scanned images are 50 μm (z-axis) in the horizontal (Fig. 3c) and 100 μm in coronal views (Supplementary Fig. 4b). The tracings of core and shell vasculature (Fig. 4bii and Supplementary Fig. 5c) were exported to Vesselucida Explorer (version 2019.1.1, MicroBrightField, Williston, VT) for branch point analysis.

**Confocal microscopy.** Images of horizontal, coronal, and sagittal SCN vasculature (Supplementary Fig. 5a) were captured on a Nikon Eclipse Ti2E confocal microscope (Nikon Inc., Melville, NY) with a 20x objective and acquired with NIS Advanced Research software (version 5.11, Nikon Inc., Melville, NY). The Z stack of the horizontal image covered 23.44 μm range with a step size of 0.97 μm. The Z stack of coronal images covered a range of 17.18 μm with a step size of 0.97 μm. The Z stack of sagittal images covered a range of 24.86 μm with a step size of 0.6 μm. The maximum intensity projections are presented.

**Statistical analysis.** To calculate branch points in the capillary network, segments of diameter greater than 10 μm were excluded[59]. The branch point is defined as a node that has three or more branches. The core and shell vasculature of 6 SCNs were used. Data were presented as mean ± standard deviation. The volume of SCN shell is $1.53 \times 10^7 \pm 1.39 \times 10^6$ μm³ and the volume of SCN core is $3.66 \times 10^6 \pm 9.16 \times 10^4$ μm³. The normality of data was confirmed by Shapiro–Wilk test on the difference of SCN shell and core branch point density ($p = 0.91$) and the two-sided paired $t$ test was conducted in R (version 4.0.0 or higher)[60] to compare the density of branch points of capillaries in the shell and core of SCN, and 0.05 α level was considered statistically significant. The figure was produced with ggplot2 (version 3.3.2 or higher)[61].

**Abbreviations.** Abbreviations of brain regions are as in the *Allen Mouse Brain Atlas*[26,28] or *The Mouse Brain in Stereotaxic Coordinates*[62].

**Reporting Summary.** Further information on research design is available in the Nature Research Reporting Summary linked to this article.

## Data availability
Data associated with this study are present in the paper or in the Supplementary Information file. The raw data that support the findings of this study are available from the corresponding authors upon reasonable request. Source data are provided with this paper.

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

## Acknowledgements

We thank Drs. K.L. Olsen and A.J. Silverman for comments on earlier drafts of the manuscript; C. McKernan for technical support; R. Tomer for advice on image processing, P. Buchanan for assistance with animal care. Light sheet microscopy was performed with support from the Zuckerman Institute's Cellular Imaging platform, and the National Institute of Health (NIH) grant 1S10OD023587-01. Confocal microscopy was performed with support from National Science Foundation (NSF) grant 1828264 to Barnard College. We are also grateful for the support of this work by National Science Foundation (NSF) grant 1749500 (to RS).

## Author contributions

Y.Y., J. L. and R.S. designed the experiment. Y.Y. and R.S. wrote the manuscript. J.L. and A.B.T. commented on early drafts of the manuscript. A.B.T. and Y.Y. acquired light-sheet microscope images and prepared figures. A.B.T. conducted confocal microscopy and light microscopy. Y.Y. conducted tissue clearing, image processing, tracing, and statistical analysis.

## Competing interests

The authors declare no competing interests.
