## [Peer Review File · Nature Communications]

Identification of the suprachiasmatic nucleus venous portal system in the mammalian brainREVIEWER COMMENTS

Reviewer #1 (Remarks to the Author):

The manuscript "The brain clock portal system: SCN-OVLT" by Yifan et al. describes a portal system connecting the hypothalamic suprachiasmatic nucleus (SCN) and the organum vasculosum of the lamina terminalis (OVLT). They presented the blood vessels of the portal system mainly based on immunostaining cleared tissues on the iDISCO methodology.

Major points:

1. Indeed, complete labeling of the brain endothelial cells with antibodies is essential in this study. Regarding immunolabeling strategy, Kirst C, et al verified that the complete vasculature was labeled combining both markers CD31 and anti-podocalyxin at the same time. However, the vasculature label strategy that the author using might not be sufficient to label complete vasculature only using type IV collagen and SMA. For example, in Figure 1 the will circle is discontinued and broken and the density level of the labeled vessel is lower compared with the previous publication.

So, a validation of effective small vessel antibody labeling is necessary in this study.

Reference:

Kirst C, Skriabine S, Vieites-Prado A, et al. Mapping the fine-scale organization and plasticity of the brain vasculature[J]. Cell, 2020, 180(4): 780-795. e25.

2. How the authors align and register to the OVLT and SCN region in the brain atlas is also a critical point. However, I didn't find this part in the method.

3. Given the data provided from light-sheet microscopy is recorded in low magnification in figure 1. It is hard to visualize the structure of the vessel connecting between SCN to OVLT. Also, the hypothesis of blood vessels communicating between SCN-OVLT need to be demonstrated in different view pictures in sup data figure 1.

4. The Authors must clearly disclose that the scope of the work is limited to the mouse brain. Otherwise, it would be an overclaim in the manuscript title. Furthermore, I suggest to add some experiments if possible and discussion about extending the discovery to the human brain.

5. The authors claimed a portal system involving the biological clock regulation. Although the structural connection between the SCN and OVLT is shown, in which way this system involves the function of the biological clock is barely investigated in the manuscript. Therefore, more functional studies of this portal system should be carried to support the conclusion the authors raised.

Minor points:

1. Author should use the same text format in the manuscript. There are nonetheless typos or odd phrases. For example, the Line 60-63 text size is 11, white text size is 12 in the manuscripts.

2. Accurately stacks stitching is necessary to prevent capillary duplications and acquire better alignment. The author should also provide the details of this part in the method.

3. The authors should cite recent literature about brain vessel labeling and tissue clearing.

Reviewer #2 (Remarks to the Author):

The paper by Yao et al. describes a novel portal system that connects the suprachiasmatic nuclei (SCN) and the organum vasculosum of the lamina terminalis (OVLT). This is an original finding that opens new possibilities for circadian regulation in the brain.

The authors were able to find this system by means of a whole-brain mounting and immunolabeling preparation (iDISCO) followed by light sheet and confocal microscopy.

The paper also examines differences between the different areas of the SCN, concluding that the shell of the nuclei exhibits a "denser and more complex vascular network" than the core area.

The images are very clear and convincing on the existence of this SCN-OVLT portal system. I only suggest some (necessary) additional remarks for the discussion of the paper.

While this is mainly descriptive study of this new humoral communication pathway, the authors could at least speculate on its neurochemical nature and function. Is this a pathway for the already known diffusible signals from the SCN (some of which were originally described by the Silver group)? If so, which is their presumptive function on the OVLT? Or could it be AVP acting as a neurohormone?

Moreover, there have been studies describing neuronal (synaptic) connections between these two areas. Trudel and Bourke (J Neurosci Methods 2003) horizontal preparation which appeared to preserve SCN-OVLT connections. The same group (Trudel and Bourke, Nature Neurosci 2010) expanded these results and showed that electrical stimulation of SCN inhibits OVLT-MNC synapses. Clock- osmoregulatory responses. Finally, Gizowsky et al. (Nature, 2016, cited in the present manuscript) repeat these findings and out them over a physiological-behavioral framework by relating sleep and thirst regulatory mechanisms. More recently, Gizowsky and Bourke (Nature, 2020) added evidence on OVLT-> SCN projections that mediate effects of osmotic changes on the circadian clock.

How is this reconciled with the current findings? Is it a dual neuronal-humoral communication scheme?

Minor

How can the authors state that the portal capillary veins lie "in the glia limitans" of the third ventricle floor?

The authors use their results as a metaphor for the Golgi-Cajal debate... which was based on neuronal, not vascular connections, so I'm not sure whether the statement applies here.

Reviewer #3 (Remarks to the Author):

In their manuscript entitled "The brain clock portal system: SCN-OVLT" Yao et al. investigated the microanatomical structure of a potential new brain/hypothalamic portal systems between the suprachiasmatic nucleus (SCN) and the vascular organ of the lamina terminalis (OVLT). For this purpose, the authors applied high resolution mapping technologies using iDISCO/free floating immunofluorescence staining and vessel tracing software tools. They visualized vessels with an overall collagen marker, arteries with smooth muscle actin, and the SCN with AVP in six brains of untreated mice. A ventral midline centered connection of microvessels/capillaries from the rostral SCN to the caudal part of microvessels/capillaries of the OVLT was revealed. The authors conclude that this describes a new (second) portal system that may convey humoral signals from the body's master clock to the OVLT and speculate that there may be more such mechanisms present in the brain for other circumventricular organs. They state that novel imaging approaches may enable the detection of these structures that may have been destroyed in the past by traditional staining approaches.

The present data is sound and nicely presented. Indeed, the detection of a new portal system in the brain is novel and will have great impact on the understanding of neurohumoral signaling in the brain/hypothalamus. The manuscript is well written and such information is of high interest for a board audience. While the SCN and the OVLT are critical for several physiological and

pathophysiological functions, pitfalls pertain to the functional proof of such signaling pathways/flow direction (with their physiological relevance), some missing information and data of previous studies that should be incorporated into the manuscript.

1) How do the authors know the direction of communication between the two brain structures? Is anything known about the direction of the flow? While functional testing may not be feasible for such small brain structures, such evidence for blood flow has previously been optioned when identifying the hypothalamic-pituitary portal system (J Physiol. 1949 May; 108(3):359-61) and should at least be discussed.

2) Some relevant previous studies that investigated the vascularization of the OVLT (and SCN) should be incorporated into the manuscript (e.g. [Vascular characteristics of the lamina terminalis of the human hypothalamus] February 1993, Medicinski Pregled 46(9-10):326-8; Morphometric analysis of the vascular network of the suprachiasmatic and paraventricular nucleus in the human brain] Article Feb 1990 Medicinski Pregled; The vascular architecture of the developing organum vasculosum of the lamina terminalis (OVLT) in the rat, Szabo, Cell Tissue Research, 1983, 233:579).

3) In particular the work by Szabo in 1983 describes some evidence for “a superficial network of the OVLT interconnected with at least three capillary beds” including (3) capillaries of the preoptic region and the retrochiasmatic area. Indeed, when looking at the pictures in this paper (Figure 5) some consistent results with the present manuscript may be present. Szabo actually already discussed a vascular connection between the choroid plexus, the subfornical organ and the OVLT but he did not know about the direction of the flow and the functional significance while showing “important structural features as neuro-haemal connections”. Such information should be discussed/incorporated into the present manuscript.

4) iDISCO staining is a powerful tool. Its limitations should be clearly acknowledged for the potential unfamiliar reader like shrinking of tissue (significant tissue shrinkage (up to 50% volume), severely impeding automated registration to the brain maps. Epilepsy Curr. 2016 Nov-Dec; 16(6): 405–407.). Did such effects play a role for the present findings?

5) The OVLT is very well known for its involvement in fever induction pathways, which should be also mentioned.

Some specific points

Abstract:

The species mouse should already be mentioned in the abstract.

It should rather read: “...hypothalamic-pituitary portal system, a structure...”

Main:

How did the authors actually identify the OVLT that is labelled in the Figures?

It should rather read “...veins, which... Here, we identified...”

Material and methods:

Humidity and ambient temperature should be added to the housing conditions of mice. An ethics approval number should be provided.

The authors have stated the manufacturer’s information for specificity of the applied primary antibodies. While the staining is convincing, adding references to previous respective controls within the same species and tissue would be preferred than manufacturer information.

Discussion:

See above

Figures:

If available, visualization of the OVLT vascular structure would be appreciated as an additional extended data file like already provided for the SCN (Extended Data Figure 3).

Reviewer #4 (Remarks to the Author):

Yao et al have investigated vascular links coursing between the suprachiasmatic nucleus and the vascular organ of the lamina terminalis (OVLT) and make the claim that a portal system linking the suprachiasmatic nucleus and OVLT exists which is a novel idea. Using state-of-the-art methodology, they have produced elegant images of straight vessels linking these two sites at the base of the preoptic/hypothalamic region. While a vascular link has been established, the authors have not yet demonstrated that this link is truly a portal system in the mode of the hypothalamus-median eminence-anterior pituitary portal system. Two important pieces of information are missing.

1. The direction of blood flow in the SCN-OVLT capillaries has not been established. This is not a trivial matter. In 1930, Popa and Fielding identified the link between the median eminence and the anterior pituitary gland, however they thought that pituitary hormones were being released into the blood stream to travel to the hypothalamus via a portal system (Popa and Fielding, *J Anatomy* 67: 227-232, 1932. It was Green and Harris (*J Physiology*, 108: 359-361, 1949) who were able, by direct vision, to establish that the portal system carried blood from the median eminence to the pituitary. At the moment, the direction of blood flow in the SCN-OVLT vessels is not clear.

2. Taking the example of the pituitary portal supply, the blood-brain barrier is not present in the median eminence allowing passage of hypothalamic releasing hormones into the portal vessels. To my knowledge, the blood-brain barrier is tight in the suprachiasmatic nucleus, so how do peptides coming from this nucleus pass into the lumen of these vessels so that they reach the OVLT?

Other points to be considered;

There is a large gap in acknowledgement of previous work on the blood supply and vascular arrangements of the OVLT. Important pioneering investigations and beautiful diagrammatic representations of H. Mergner (*Affen. Z. wiss. Zool.* 165: 140-185, 1961); Henri M. Duvernoy and colleagues (summarised in *Brain Res Rev* 56: 119-147, 2007) and Adolf Weindl (*Grafe and Weindl, Wiss Z. Karl-Marx-Univ Leipzig, Math.-Naturwiss. R.* 36(2): 214-220, 1987) should be mentioned. Indeed, the latter publication shows blood vessels emanating ventrally and caudally from the OVLT around the surface of the optic chiasm to caudal sites. This publication also discusses the idea of a portal system from the OVLT to other brain sites, but comes out against this idea.

There is no mention of the species studied until line 278. As well, there is no indication of how many animals were investigated.

Were appropriate controls to determine the specificity of the AVP immunohistochemistry performed (i.e. was preabsorption of the antisera, or leaving out the primary antibodies checked?). Abbreviations SCN and OVLT should not be used in the title.

May 19, 2021 revised June 9 2021
Ms. # NCOMMS-20-47413-T

Thank you for the review of our manuscript. We appreciate the thoughtful comments of the four reviewers. The questions and comments have led us to expand the documentation of our major finding - namely the establishment of a portal system between the suprachiasmatic nucleus (SCN) and the organum vasculosum of the lamina terminalis (OVLT).

We alert the reviewers to the fact that this manuscript was initially submitted to Nature, and complied with Nature's very short 2000 word limit format. This very severely limited our opportunities to include relevant information. We have now redirected the manuscript to Nature Communications and have expanded it in keeping with the suggestions of the reviewers.

We present comment in turn in normal font and provide our responses in **red font**. The changes made in the manuscript also shown are in **red font**.

Reviewers' remarks:

Reviewer #1: (Remarks to the Author)

The manuscript "The brain clock portal system: SCN-OVLT" by Yifan et al. describes a portal system connecting the hypothalamic suprachiasmatic nucleus (SCN) and the organum vasculosum of the lamina terminalis (OVLT). They presented the blood vessels of the portal system mainly based on immunostaining cleared tissues on the iDISCO methodology.

Major points:

1. Indeed, complete labeling of the brain endothelial cells with antibodies is essential in this study. Regarding immunolabeling strategy, Kirst C, etc. verified that the complete vasculature was labeled combining both markers CD31 and anti-podocalyxin at the same time. However, the vasculature label strategy that the author using might not be sufficient to label complete vasculature only using type IV collagen and SMA. For example, in Figure 1 the will circle is discontinued and broken and the density level of the labeled vessel is lower compared with the previous publication. So, a validation of effective small vessel antibody labeling is necessary in this study.

We are familiar with this excellent paper. While Kirst et al. explored whole brain vasculature, in their paper they do not show any images of our areas of the SCN, or the OVLT, or any of the other CVOs¹. In contrast, we have focused on these regions. We note that our results provide higher magnification images than those available in the Kirst paper. Most importantly, the collagen marker we used is a component of all blood vessels². Kirst used a cocktail of three antibodies (CD31, podocalyxin, and Acta2), and while this is an excellent strategy, it is not necessarily proof that all vessels were stained. Neither of our methods can exclude the possibility that some blood vessels were missed. The reviewer notes that the use of additional protocols may reveal more data regarding capillary vessels. This is a possibility, given that one

cannot prove the negative. Whether or not some blood vessels were missed, our results demonstrate the presence of portal vessels between SCN and OVLT.

Kirst et al in Table S1 provide a statistic for branching density of sensory CVO's¹. To compare our results to theirs we recalculated the branching density of the whole SCN as in Kirst et al. (which is 25 μ m voxel). The result is 0.132 ± 0.008 per unit volume. This result is very close to the calculation from Kirst et al., that is 0.138 ± 0.029 per unit volume. This information has been incorporated into the results. However, our data also show subregional analysis within the SCN as displayed in Figure 3. This can't be achieved by brain registration method by Kirst et al.

2. How the authors align and register to the OVLT and SCN region in the brain atlas is also a critical point. However, I didn't find this part in the method.

We did not need to register the cleared material to the brain atlas. The SCN and OVLT were identified in the same single scan by the method we used. The only comparison we made was in estimating the distance travelled by portal vessels. This was $\sim 385\mu$ m in our material and estimated at $\sim 425\mu$ m in the Allen Brain Atlas. This point is now clarified in the text that we did not need to register our material to a standard.

3. Given the data provided from light-sheet microscopy is recorded in low magnification in figure 1. It is hard to visualize the structure of the vessel connecting between SCN to OVLT. Also, the hypothesis of blood vessels communicating between SCN-OVLT need to be demonstrated in different view pictures in supplementary data Figure 1.

We thank the reviewer for this comment. As requested, we now provide additional views of the vascular connections (Fig. 2, Supplementary Fig. 3, and Supplementary Video 1). In the revision, we have provided additional orientations and a video. These clearly show the continuity of the portal vessels coursing between the SCN and the OVLT. Note that the figure in the initial submission (which is retained) emphasizes the bilaterally symmetrical nature of the vasculature of the bilateral suprachiasmatic nucleus.

4. The Authors must clearly disclose that the scope of the work is limited to the mouse brain. Otherwise, it would be an overclaim in the manuscript title. Furthermore, I suggest to add some experiments if possible and discussion about extending the discovery to the human brain.

Thank you for this suggestion. We now indicate in a more salient way that the work was done in mice. It certainly would be interesting to have data on the human brain, and we note that CVOs are highly conserved in mammals including humans.

5. The authors claimed a portal system involving the biological clock regulation. Although the structural connection between the SCN and OVLT is shown, in which way this system involves the function of the biological clock is barely investigated in the manuscript. Therefore, more functional studies of this portal system should be carried to support the conclusion the authors raised.

The possible function of this new portal system and the well-established functions of the SCN and the OVLT are now covered in the discussion.

Minor points:

1. Author should use the same text format in the manuscript. There are nonetheless typos or odd phrases. For example, the Line 60-63 text size is 11, white text size is 12 in the manuscripts.

This is fixed.

2. Accurately stacks stitching is necessary to prevent capillary duplications and acquire better alignment. The author should also provide the details of this part in the method.

As noted above, this study did not use stitching. The following explanation is now provided in the text: The xy-plane of the higher power image of the SCN and the OVLT region is 1390 μm x 640 μm for the sagittal scan (Fig. 2; Supplementary Fig. 3) and 730 μm x 1000 μm for the horizontal scan (Fig. 3; Supplementary Fig. 4). The light sheet microscope mounted with a 12x 0.53NA objective has a field of view (FOV) as 1.39mm X 1.17mm, which is larger than the image tile we used for analysis. The lower power image we showed as Fig. 1 was taken with 1.3x 0.1NA with a FOV of 10.8mm x 12.8mm. This FOV is larger than the xy dimension of Figure 1 (6.4mm x 9.7mm). Thus, one tile covered regions of interest in the light sheet microscopy in the current study.

3. The authors should cite recent literature about brain vessel labeling and tissue clearing.

This has been done.

Reviewer #2:

The paper by Yao et al. describes a novel portal system that connects the suprachiasmatic nuclei (SCN) and the organum vasculosum of the lamina terminalis (OVLT). This is an original finding that opens new possibilities for circadian regulation in the brain. The authors were able to find this system by means of a whole-brain mounting and immunolabeling preparation (iDISCO) followed by light sheet and confocal microscopy. The paper also examines differences between the different areas of the SCN, concluding that the shell of the nuclei exhibits a “denser and more complex vascular network” than the core area. The images are very clear and convincing on the existence of this SCN-OVLT portal system. I only suggest some (necessary) additional remarks for the discussion of the paper.

We thank the reviewer for these positive comments.

While this is mainly descriptive study of this new humoral communication pathway, the authors could at least speculate on its neurochemical nature and function. Is this a pathway for the already known diffusible signals from the SCN (some of which were originally described by the Silver group)? If so, which is their presumptive function on the OVLT? Or could it be AVP acting as a neurohormone? Moreover, there have been studies describing neuronal (synaptic) connections between these two areas. Trudel and Bourke (J Neurosci Methods 2003) horizontal preparation which appeared to preserve SCN-OVLT connections. The same group

(Trudel and Bourke, Nature Neurosci 2010) expanded these results and showed that electrical stimulation of SCN inhibits OVLT-MNC synapses. Clock- osmoregulatory responses. Finally, Gizowsky et al. (Nature, 2016, cited in the present manuscript) repeat these findings and out them over a physiological-behavioral framework by relating sleep and thirst regulatory mechanisms. More recently, Gizowsky and Bourke (Nature, 2020) added evidence on OVLT-> SCN projections that mediate effects of osmotic changes on the circadian clock. How is this reconciled with the current findings? Is it a dual neuronal-humoral communication scheme? **The work of Bourque is well known to us and much admired. The Bourque studies (and others), pointing to neural connections between SCN and OVLT were previously cited. As suggested, we now mention possible functions of a dual neuronal-humoral communication scheme between the SCN and OVLT.**

Minor

How can the authors state that the portal capillary veins lie “in the glia limitans” of the third ventricle floor?

We now indicate that the vessels lie along the floor of the 3rd ventricle (without specifying their location with regard to the glia limitans).

The authors use their results as a metaphor for the Golgi-Cajal debate... which was based on neuronal, not vascular connections, so I'm not sure whether the statement applies here.

We have rewritten this comment to state clearly that we are using it as a conceptually interesting metaphor.

Reviewer #3 (Remarks to the Author):

In their manuscript entitled "The brain clock portal system: SCN-OVLT" Yao et al. investigated the microanatomical structure of a potential new brain/hypothalamic portal systems between the suprachiasmatic nucleus (SCN) and the vascular organ of the lamina terminalis (OVLT). For this purpose, the authors applied high resolution mapping technologies using iDISCO/free floating immunofluorescence staining and vessel tracing software tools. They visualized vessels with an overall collagen marker, arteries with smooth muscle actin, and the SCN with AVP in six brains of untreated mice. A ventral midline centered connection of microvessels/capillaries from the rostral SCN to the caudal part of microvessels/capillaries of the OVLT was revealed. The authors conclude that this describes a new (second) portal system that may convey humoral signals from the body's master clock to the OVLT and speculate that there may be more such mechanisms present in the brain for other circumventricular organs. They state that novel imaging approaches may enable the detection of these structures that may have been destroyed in the past by traditional staining approaches.

The present data is sound and nicely presented. Indeed, the detection of a new portal system in the brain is novel and will have great impact on the understanding of neurohumoral signaling in the brain/hypothalamus. The manuscript is well written and such information is of high interest for a board audience. While the SCN and the OVLT are critical for several physiological and pathophysiological functions, pitfalls pertain to the functional proof of such signaling

pathways/flow direction (with their physiological relevance), some missing information and data of previous studies that should be incorporated into the manuscript.

We thank the reviewer for the favourable comments. The new anatomical data in our work provides a rationale for future functional studies that would not be considered in the absence of such data.

1. How do the authors know the direction of communication between the two brain structures? Is anything known about the direction of the flow? While functional testing may not be feasible for such small brain structures, such evidence for blood flow has previously been obtained when identifying the hypothalamic-pituitary portal system (Green and Harris, J Physiol. 1949 May; 108(3):359-61) and should at least be discussed.

Demonstration of the direction of blood flow is a very reasonable, logical and necessary **next** question. Determination of the direction blood flow in this brain region in the mouse (or rat) will entail many years of work to establish. We can imagine other ways of approaching the problem in mice or rats, but these represent several years of work. We suggest that this is not a reasonable requirement for the publication of the present study.

As the reviewers know, direct confirmation of the direction of blood flow in this system is a herculean task. Reviewer #4 referred us to the reference of Green and Harris³ who wrote “direct observation of the direction of blood flow in the living animal.” Furthermore, Green and Harris state that in studies of the pituitary portal system, “Rats were used since they possess long hypophysiportal vessels, in a horizontal plane, which are readily approached and easily observed.” The SCN-OVLT portal vessels are neither “readily approached nor easily observed”. The vessels are short (<400µm) and lie above the optic chiasm on the floor of the 3rd ventricle above the bone at the base of the brain. Because the SCN and OVLT lie above the optic chiasm, even if the bone were chipped away as in Green and Harris, it would still leave the near-impossible task of removing the optic chiasm to visualize blood vessels, while keeping intact the thin membrane at the floor of the 3rd ventricle.

Perhaps a preparation in a large animal such as sheep would be tractable, but beyond feasible in our laboratory. That said, the SCN and OVLT are very highly conserved, and our findings are likely to stimulate renewed interest among neuroendocrinologists and circadian biologists.

We cite the publication of Harris rather than the very brief paper by Green and Harris (1949)³ as the latter does not show any of experimental results and the paper by Harris (1948)⁴ is published earlier.

2. Some relevant previous studies that investigated the vascularization of the OVLT (and SCN) should be incorporated into the manuscript (e.g. Vascular characteristics of the lamina terminalis of the human hypothalamus, Feb 1993, Medicinski Pregled 46(9-10):326-8⁵; Morphometric analysis of the vascular network of the suprachiasmatic and paraventricular nucleus in the human brain, Feb 1990 Medicinski Pregled⁶; The vascular architecture of the

developing organum vasculosum of the lamina terminalis (OVLT) in the rat, Szabo, Cell Tissue Research, 1983, 233:579⁷).

The goal of this paper is the demonstration of the blood vessels joining two capillary beds, thereby forming a portal system rather than exploring the capillary beds within each of these nuclei. There is no prior work on this portal system. As the reviewer rightfully points out, the vascularization of the OVLT has been thoroughly studied. That of the SCN somewhat less. As suggested, we now cite the vascularity work of Szabo (1983)⁷ regarding OVLT and that of Ambach and Palkovitz (1974)⁸ regarding SCN, but point out that they are not reporting in a portal system. We successfully obtained and translated the Croatian papers in Medicinski Pregled from a colleague in Poland and cite their work on human SCN vasculature⁶.

3. In particular the work by Szabo in 1983 describes some evidence for “a superficial network of the OVLT interconnected with at least three capillary beds” including (3) capillaries of the preoptic region and the retrochiasmatic area. Indeed, when looking at the pictures in this paper (Figure 5) some consistent results with the present manuscript may be present. Szabo actually already discussed a vascular connection between the choroid plexus, the subfornical organ and the OVLT but he did not know about the direction of the flow and the functional significance while showing “important structural features as neuro-haemal connections”. Such information should be discussed/incorporated into the present manuscript.

The SCN is not part of Szabo’s study material and he did not examine the connection between the SCN and OVLT. As noted, Szabo suggested a possible vascular connection among CVO’s (choroid plexus, the subfornical organ and the OVLT) and this is now cited.

4. iDISCO staining is a powerful tool. Its limitations should be clearly acknowledged for the potential unfamiliar reader like shrinking of tissue (significant tissue shrinkage (up to 50% volume), severely impeding automated registration to the brain maps¹⁰. Did such effects play a role for the present findings?

We now clarify that we did not need to do registration as all regions of interest were within the field of view in our scan, so this problem does not apply. The protocol we used produces shrinkage of about 11% and this is now mentioned in the text. The xy-plane of the higher power image of the SCN and the OVLT region is 1390 μm x 640 μm for the sagittal scan (Fig. 2; Supplementary Fig. 3) and 730 μm x 1000 μm for the horizontal scan (Fig. 3; Supplementary Fig. 4). The light sheet microscope mounted with a 12x 0.53NA objective has a field of view (FOV) as 1.39mm X 1.17mm, which is larger than the image tile we used for analysis. The lower power image we showed as Fig. 1 was taken with 1.3x 0.1NA with a FOV of 10.8mm x 12.8mm. This FOV is larger than the xy dimension of Figure 1 (6.4mm x 9.7mm). Thus, one tile covered regions of interest in the light sheet microscopy in the current study.

5. The OVLT is very well known for its involvement in fever induction pathways, which should be also mentioned.

As requested, we mention fever induction and additional functions of OVLT and the possible involvement of the SCN- OVLT portal pathway in these responses.

Some specific points

Abstract:

The species mouse should already be mentioned in the abstract.

It should rather read: "...hypothalamic-pituitary portal system, a structure..."

This has been done.

Main:

How did the authors actually identify the OVLT that is labelled in the Figures?

The OVLT is readily recognized in this material by its collagen-labelled vasculature (Fig. 2e).

It should rather read "...veins, which... Here, we identified..."

This is corrected. Thank you.

Material and methods:

Humidity and ambient temperature should be added to the housing conditions of mice. An ethics approval number should be provided.

This has been done as suggested.

The authors have stated the manufacturer's information for specificity of the applied primary antibodies. While the staining is convincing, adding references to previous respective controls within the same species and tissue would be preferred than manufacturer information.

This is now provided in methods. We performed a no primary control and a collagen dilution series to optimize staining. We also tested for optimal dilutions for the 3 antibodies (collagen, SMA, AVP) used in triple label studies in cleared iDisco material.

Figures:

If available, visualization of the OVLT vascular structure would be appreciated as an additional extended data file like already provided for the SCN

We added higher magnification images as requested in Figure 2e. We also provide additional views of the vascular connections (Fig. 2, Supplementary Fig. 3, and Supplementary Video 1), as requested. Note that the initial submission emphasized the vasculature on both sides of the bilateral suprachiasmatic nucleus. In the revision, we have provided additional orientations and a video. These clearly show the continuity of the portal vessels coursing between the SCN and the OVLT.

Reviewer #4

Yao et al have investigated vascular links coursing between the suprachiasmatic nucleus and the vascular organ of the lamina terminalis (OVLT) and make the claim that a portal system linking the suprachiasmatic nucleus and OVLT exists which is a novel idea. Using state-of-the-art methodology, they have produced elegant images of straight vessels linking these two sites at the base of the preoptic/hypothalamic region. While a vascular link has been established, the

authors have not yet demonstrated that this link is truly a portal system in the mode of the hypothalamus-median eminence-anterior pituitary portal system. Two important pieces of information are missing.

1. The direction of blood flow in the SCN-OVLT capillaries has not been established. This is not a trivial matter. In 1930, Popa and Fielding⁹ identified the link between the median eminence and the anterior pituitary gland, however they thought that pituitary hormones were being released into the blood stream to travel to the hypothalamus via a portal system⁹. It was Green and Harris³ who were able, by direct vision, to establish that the portal system carried blood from the median eminence to the pituitary. At the moment, the direction of blood flow in the SCN-OVLT vessels is not clear.

This has been addressed in the response to Reviewer #3 and is repeated in part here: Demonstration of the direction of blood flow is a very reasonable, logical and necessary **next** question. Determination of the direction blood flow in this brain region in the mouse (or rat) will entail many years of work to establish. We can imagine ways of approaching the problem in mice or rats, but these represent several years of work. We suggest that this is not a reasonable requirement for the publication of the present study.

As the researchers know, direct confirmation of the direction of blood flow in this portal system will be a herculean task. This reviewer referred us to the reference of Green and Harris³ who wrote “direct observation of the direction of blood flow in the living animal.” Furthermore, Green and Harris state that in studies of the pituitary portal system, “Rats were used since they possess long hypophysoportal vessels, in a horizontal plane, which are readily approached and easily observed.” The SCN-OVLT portal vessels are neither “readily approached nor easily observed”. The vessels are short (<400um) and lie above the optic chiasm on the floor of the 3rd ventricle above the bone at the base of the brain. Because the SCN and OVLT lie above the optic chiasm, even if the bone were removed as in Green and Harris, it would still leave the near-impossible task of removing the optic chiasm without damaging the floor of the 3rd ventricle so as to visualize blood vessels.

2. Taking the example of the pituitary portal supply, the blood-brain barrier is not present in the median eminence allowing passage of hypothalamic releasing hormones into the portal vessels. To my knowledge, the blood-brain barrier is tight in the suprachiasmatic nucleus, so how do peptides coming from this nucleus pass into the lumen of these vessels so that they reach the OVLT?

We thank the reviewer for raising this important point, and the relevant information has now been incorporated into the manuscript. The reviewer is correct in saying that the SCN, like other parenchymal regions has an intact blood-brain barrier (BBB). With regard to brain to vasculature communication, there is a substantial literature documenting the ability of peptides, including vasopressin, to cross the BBB in the parenchyma in the presence of tight junctions in blood vessels.

Other points to be considered;

There is a large gap in acknowledgement of previous work on the blood supply and vascular arrangements of the OVLT. Important pioneering investigations and beautiful diagrammatic representations of H. Mergner¹⁰, Henri M. Duvernoy and colleagues¹¹ and Adolf Weindl¹² should be mentioned. Indeed, the latter publication shows blood vessels emanating ventrally and caudally from the OVLT around the surface of the optic chiasm to caudal sites. This publication also discusses the idea of a portal system from the OVLT to other brain sites, but comes out against this idea.

We concur that the vascularization of the OVLT has been thoroughly studied. It was not our goal to study the vascularization of each CVO. The goal of the paper is to demonstrate that the capillary beds of SCN and OVLT are joined, thereby forming a portal system. Given its relevance to the present study, we have provided additional figures showing the vasculature of the OVLT and the site of entry of the portal vessels.

For the second point, (Grafe and Weindl, ...“ publication also discusses the idea of a portal system from the OVLT to other brain sites, but comes out against this idea”)¹², we thank the reviewer for this comment and now incorporate this reference. Importantly, the authors did not study the OVLT-SCN connection. They only examined OVLT -POA and concluded that a portal system was lacking at *that* site.

There is no mention of the species studied until line 278. As well, there is no indication of how many animals were investigated.

Thank you. This is now corrected. In addition, we indicate numbers of animals used in the study.

Were appropriate controls to determine the specificity of the AVP immunohistochemistry performed (i.e., was preabsorption of the antisera, to leaving out the primary antibodies checked?).

Reviewer #3 also raised this point. We performed a no primary and collagen dilution series to optimize staining. We also tested for optimal dilutions for the 3 antibodies (collagen, SMA, AVP) used in triple label studies in cleared iDISCO material. This information is now provided in methods.

References

- 1 Kirst, C. *et al.* Mapping the Fine-Scale Organization and Plasticity of the Brain Vasculature. *Cell* **180**, 780-795.e725, (2020).
- 2 Urabe, N. *et al.* Basement Membrane Type IV Collagen Molecules in the Choroid Plexus, Pia Mater and Capillaries in the Mouse Brain. *Archives of Histology and Cytology* **65**, 133-143, (2002).
- 3 Green, J. D. & Harris, G. W. Observation of the hypophysiportal vessels of the living rat. **108**, 359-361, (1949).
- 4 Harris, G. J. P. R. Neural control of the pituitary gland. **28**, 139-179, (1948).
- 5 Polzović, A., Mihić, N. & Cvejin, B. Vascular characteristics of the lamina terminalis of the human hypothalamus. *Medicinski preglad* **46**, 326-328, (1993).

- 6 Polzović, A., Mihić, N. & Mijatov-Ukropina, L. [Morphometric analysis of the vascular network of the suprachiasmatic and paraventricular nucleus in the human brain]. *Medicinski preglad* **43**, 136-139, (1990).
- 7 Szabó, K. The vascular architecture of the developing organum vasculosum of the lamina terminalis (OVLT) in the rat. *Cell and Tissue Research* **233**, 579-592, (1983).
- 8 Ambach, G. & Palkovits, M. Blood supply of the rat hypothalamus. III. Anterior region of the hypothalamus (nucleus suprachiasmaticus, nucleus hypothalamicus anterior, nucleus periventricularis). *Acta morphologica Academiae Scientiarum Hungaricae* **23**, 21-49, (1975).
- 9 Popa, G. T. & Fielding, U. Hypophysio-Portal Vessels and their Colloid Accompaniment. *J Anat* **67**, 227-232.221, (1933).
- 10 Mergner, H. J. Z. w. Z. Die Blutversorgung der Lamina terminalis bei einigen Affen. **165**, 140-185, (1961).
- 11 Duvernoy, H. M. & Risold, P.-Y. The circumventricular organs: An atlas of comparative anatomy and vascularization. *Brain Research Reviews* **56**, 119-147, (2007).
- 12 Grafé, G. & Weindl, A. J. W. Z. K. M.-U. L., Math-Naturwiss. The vascular connections of the organum vasculosum of the lamina terminalis in the rat. **36**, 214-220, (1987).

REVIEWER COMMENTS

Reviewer #1 (Remarks to the Author):

I appreciate authors work to address reviewer comments.

I have some minor comments that could improve manuscript further.

1. As the authors only provide the possible structure of the portal system in the manuscript, it still lacks direct evidence of the secretion of the specialized target between SCN and OVLT via the portal system. It might go into the general circulatory system via other capillary branches. Authors could discuss/comment on that.

2. Moreover, the Authors use the same age and gender mice in this study, claiming a new finding of a portal system in the brain. A staining of the portal system covering different strains or gender of mice could ave been worth. Authors could discuss/comment on that.

Reviewer #2 (Remarks to the Author):

I have carefully read the new manuscript and the replies to all four reviewers, and consider that he manuscript has been greatly improved and all concerns have been covered.

However, some of the additions to the text might be considered repetitive (for example, when over-explaining the rationale in te results section).

As stated in my original review I maintain that this paper will be a fundamental addition to the existing literature on SCN outputs (and/or inputs, depending on the direction of the flow to be uncovered by further studies).

Reviewer #3 (Remarks to the Author):

While the authors were not able to provide any functional evidence, the authors have added very helpful additional information, which better conveys their findings to the reader. Overall, this referee agrees that such information remains important to answer logical and next questions in the future like the direction of blood flow. Thus, the authors should clearly state limitations for more clarity to the reader. Otherwise, the authors appropriately addressed the comments of this referee.

Reviewer #4 (Remarks to the Author):

This revised manuscript gives a superb description of the straight vascular connections between the suprachiasmatic nucleus and OVLT of the mouse. The authors have largely addressed the concerns of the reviewers. However, I consider that until the direction of blood-flow in these vessels is established, the authors should soften their conclusion which largely only considers that this vascular connection between SCN and OVLT is a portal system from SCN to OVLT. While they state now that the work "does not prove that the direction of communication is from the brain clock to OVLT", it should state explicitly that blood flow direction needs to be established. It is also possible that neuroendocrine secretion in OVLT diffuses into a hypothetical OVLT to SCN portal system. For example, there is a rich projection of preoptic LHRH-secreting fibers into the OVLT in the vicinity of its superficial capillary plexus that is similar to that in the median eminence (see McKinley et al. *Frontiers Neuroendocrinol.* 11:91-127, 1990). I consider that the authors should explicitly state that this vascular connection between OVLT and SCN could represent either an SCN to OVLT portal system and/or an OVLT to SCN portal connection. As well, the new sentences (line 205-211) is not convincing because the various functions e.g. thirst, ovulation, osmoregulation, fever can be influenced at a number of other preoptic/hypothalamic sites other than the OVLT.

June 29, 2021
Ms. # NCOMMS-20-47413B

Dear anonymous reviewers,

Thank you for your reviews of our manuscript. We appreciate the careful comments and find them helpful in sharpening the discussion on the portal system between the suprachiasmatic nucleus (SCN) - organum vasculosum of the lamina Terminalis (OVLT) and its possible functionality.

We present each comment in turn in black font and provide our responses in **red font**. The changes made in the manuscript also shown in **red font**.

Reviewer #1 (Remarks to the Author):

I appreciate authors work to address reviewer comments.

I have some minor comments that could improve manuscript further.

1. As the authors only provide the possible structure of the portal system in the manuscript, it still lacks direct evidence of the secretion of the specialized target between SCN and OVLT via the portal system. It might go into the general circulatory system via other capillary branches. Authors could discuss/comment on that.

We concur that signals from the SCN may reach the general circulatory system and have added that point to the discussion.

2. Moreover, the Authors use the same age and gender mice in this study, claiming a new finding of a portal system in the brain. A staining of the portal system covering different strains or gender of mice could have been worth. Authors could discuss/comment on that.

We added a comment on the need to examine additional species, the point that more strains and ages of mice should be examined as well.

Reviewer #2 (Remarks to the Author):

I have carefully read the new manuscript and the replies to all four reviewers, and consider that the manuscript has been greatly improved and all concerns have been covered.

However, some of the additions to the text might be considered repetitive (for example, when over-explaining the rationale in the results section).

We have now shortened the rationale in the results section, as requested here.

We note that the rationale is laid out in detail in the results section as this was questioned in the previous review. Specifically, a previous review questioned our ability to register the tissue against a benchmark, pointed to the possibility of error arising from tissue shrinkage, and asked about the limitations of our staining protocols. The careful explanation in the results sections resolved these issues.

As stated in my original review I maintain that this paper will be a fundamental addition to the existing literature on SCN outputs (and/or inputs, depending on the direction of the flow to be uncovered by further studies).

We thank the reviewer for this comment.

Reviewer #3 (Remarks to the Author):

While the authors were not able to provide any functional evidence, the authors have added very helpful additional information, which better conveys their findings to the reader. Overall, this referee agrees that such information remains important to answer logical and next questions in the future like the direction of blood flow. Thus, the authors should clearly state limitations for more clarity to the reader. Otherwise, the authors appropriately addressed the comments of this referee.

The discussion point on important next questions in the future like the direction of blood flow has been sharpened by stating directly that the direction of blood flow needs to be established.

Reviewer #4 (Remarks to the Author):

This revised manuscript gives a superb description of the straight vascular connections between the suprachiasmatic nucleus and OVLT of the mouse. The authors have largely addressed the concerns of the reviewers. However, I consider that until the direction of blood-flow in these vessels is established, the authors should soften their conclusion which largely only considers that this vascular connection between SCN and OVLT is a portal system from SCN to OVLT. While they state now that the work "does not prove that the direction of communication is from the brain clock to OVLT", it should state explicitly that blood flow direction needs to be established.

We have adjusted the wording to state explicitly that blood flow direction needs to be established rather than the wording previously used (which stated that the research did not prove the direction of flow).

It is also possible that neuroendocrine secretion in OVLT diffuses into a hypothetical OVLT to SCN portal system. For example, there is a rich projection of preoptic LHRH-secreting fibers into the OVLT in the vicinity of its superficial capillary plexus that is similar to that in the median eminence (see McKinley et al. *Frontiers Neuroendocrinol.* 11:91-127, 1990). I consider that the authors should explicitly state that this vascular connection between OVLT and SCN could represent either an SCN to OVLT portal system and/or an OVLT to SCN portal connection.

We now state explicitly that the vascular connection between OVLT and SCN could represent either an SCN to OVLT portal system and/or an OVLT to SCN portal connection.

As well, the new sentences (line 205-211) is not convincing because the various functions e.g. thirst, ovulation, osmoregulation, fever can be influenced at a number of other preoptic/hypothalamic sites other than the OVLT.

We fully concur and add the point that these functions can be influenced at a number of other preoptic/hypothalamic sites other than the OVLT.

REVIEWERS' COMMENTS

Reviewer #1 (Remarks to the Author):

I dont have further comments.

Reviewer #3 (Remarks to the Author):

Limitations are now more clearly mentioned; the authors appropriately addressed the remaining concern of this referee.

Reviewer #4 (Remarks to the Author):

The authors have responded to the reviewer's comments and revised the manuscript appropriately. It makes a significant contribution to knowledge of hypothalamic function.